# Calcium and Neural Stem Cell Proliferation

**DOI:** 10.3390/ijms25074073

**Published:** 2024-04-06

**Authors:** Dafne Astrid Díaz-Piña, Nayeli Rivera-Ramírez, Guadalupe García-López, Néstor Fabián Díaz, Anayansi Molina-Hernández

**Affiliations:** 1Departamento de Fisiología y Desarrollo Celular, Instituto Nacional de Perinatología Isidro Espinosa de los Reyes, Montes Urales 800, Miguel Hidalgo, Ciudad de México 11000, Mexico; astrid.diaz@hotmail.com (D.A.D.-P.); rirn.dox@gmail.com (N.R.-R.); guadalupegl2000@yahoo.com.mx (G.G.-L.); nfdiaz00@yahoo.com.mx (N.F.D.); 2Facultad de Medicina, Circuito Exterior Universitario, Universidad Nacional Autónoma de México Universitario, Copilco Universidad, Coyoacán, Ciudad de México 04360, Mexico

**Keywords:** calcium signaling, neural stem/progenitor cells, proliferation, differentiation, radial glial cells

## Abstract

Intracellular calcium plays a pivotal role in central nervous system (CNS) development by regulating various processes such as cell proliferation, migration, differentiation, and maturation. However, understanding the involvement of calcium (Ca^2+^) in these processes during CNS development is challenging due to the dynamic nature of this cation and the evolving cell populations during development. While Ca^2+^ transient patterns have been observed in specific cell processes and molecules responsible for Ca^2+^ homeostasis have been identified in excitable and non-excitable cells, further research into Ca^2+^ dynamics and the underlying mechanisms in neural stem cells (NSCs) is required. This review focuses on molecules involved in Ca^2+^ entrance expressed in NSCs in vivo and in vitro, which are crucial for Ca^2+^ dynamics and signaling. It also discusses how these molecules might play a key role in balancing cell proliferation for self-renewal or promoting differentiation. These processes are finely regulated in a time-dependent manner throughout brain development, influenced by extrinsic and intrinsic factors that directly or indirectly modulate Ca^2+^ dynamics. Furthermore, this review addresses the potential implications of understanding Ca^2+^ dynamics in NSCs for treating neurological disorders. Despite significant progress in this field, unraveling the elements contributing to Ca^2+^ intracellular dynamics in cell proliferation remains a challenging puzzle that requires further investigation.

## 1. Introduction

In mammals, the central nervous system (CNS) originates from a simple neuroepithelium populated by neural stem cells (NSCs), known as neuroepithelial cells. Through symmetric proliferation, these cells give rise to a pseudo-stratified neuroepithelium, where bipolar NSCs called radial glia cells (RGCs) are observed. RGCs are multipotent stem cells capable of self-renewal and differentiation into neurons, astrocytes, and oligodendrocytes in a time-dependent manner through symmetric and asymmetric cell division [1,2,3].

NSCs, also referred to as neural progenitor cells (NPCs), must be distinguished from intermediate progenitor cells (IPCs), as the latter are committed to a neuronal phenotype. NSCs and IPCs constitute two proliferative zones in the ventricular and subventricular zones (VZ and SVZ), respectively. IPCs primarily generate pyramidal neurons of the cerebral cortex, while RGCs directly produce a small portion of neurons, astrocytes, and oligodendrocytes [4,5,6,7].

Both extrinsic and intrinsic factors regulate NSCs’ self-renewal and differentiation in time and space, thereby affecting cell proliferative patterns, partly determined based on cell cycle duration [8,9]. Extrinsic factors such as growth factors, neurotransmitters, and morphogens play significant roles in CNS development through selective membrane receptors. Receptor tyrosine kinases (RTKs), guanine nucleotide-binding protein-coupled receptors (G-protein-coupled receptors; GPCRs), and mechanoreceptors are among the receptors capable of eliciting intracellular calcium (Ca^2+^) transients in non-excitable cells. Consequently, the increase in intracellular Ca^2+^ in NSC is mediated downstream from receptor activation via two mechanisms: release from store organelles and entry through plasma membrane non-voltage-dependent channels.

The activity of Ca^2+^ as a second messenger relies on changes in the intracellular Ca^2+^ concentration ([Ca^2+^]i) at microdomains sensed by several Ca^2+^ binding proteins that regulate intracellular Ca^2+^ homeostasis by buffering and transporting this ion. These proteins include calmodulin (CaM), calcyclin, Ca^2+^-sensitive enzymes, transcription factors, and Ca^2+^-sensitive ion channels. Additionally, pumps and exchangers localized in mitochondria, endoplasmic reticulum (ER), and Golgi membranes contribute to Ca^2+^ homeostasis [10].

Due to the wide molecular repertoire involved in Ca^2+^ homeostasis, it is challenging to associate the molecules involved in Ca^2+^ dynamics and cell proliferation in the context of CNS development, as several phenomena coincide in time and space. However, proliferation is the first event during neural tube development, and alterations in this process can lead to CNS defects such as microcephaly, megalocephaly, and hemimegaloencephaly [11].

In the VZ, where NSCs reside in the cortical neuroepithelium of rat embryos, three patterns of spontaneous Ca^2+^ fluctuations have been reported: (1) isolated fluctuation in a single cell, (2) paired fluctuation in two adjacent cells, and (3) clustered or synchronized fluctuation in neighboring cells, each with particular kinetics and frequency patterns [12,13]. Interestingly, in 15- to 20-day-old rat embryos (E15–E20), most cells in the VZ exhibit the isolated Ca^2+^ fluctuation pattern, followed by pair fluctuation and cluster fluctuations [13]. However, these patterns have not been related to symmetric or asymmetric RGC proliferation or the expression of non-voltage-gated channels.

NSCs proliferate symmetrically to increase their pool or asymmetrically to give rise, directly or indirectly, in a time-dependent manner, to neurons, astrocytes, and oligodendrocytes (Figure 1). The potential fates of NSCs rely on their proliferative pattern [8]. Furthermore, symmetric cell division in the VZ is more frequent at early stages, and as development progresses, the asymmetric pattern starts to generate committed cells with or without proliferative capacity [14,15,16].

The proliferation and differentiation of NSC into neurons, astrocytes, and oligodendrocytes is a complex and highly regulated process. Certain factors exhibit dual roles depending on their concentration and cellular context. Among these factors are Notch, bone morphogenetic proteins (BMPs), brain-derived neurotrophic factor (BDNF), sonic hedgehog (SHH), Wingless (Wnt), and nerve growth factor (NGF).

Epidermal growth factor (EGF), fibroblast growth factor (FGF), and insulin-like growth factor (IGF) promote NSC proliferation. Neurogenic factors include neurotrophin-3 (NT-3), vascular endothelial growth factor (VEGF), and transforming growth factor-beta (TGF-β). Astrocyte differentiation is influenced by ciliary neurotrophic factor (CNTF), BMP2, BMP4, leukemia inhibitory factor (LIF), and interleukin-6 (IL-6). Oligodendrocyte differentiation, on the other hand, is influenced by platelet-derived growth factor (PDGF), IGF-I, neuregulin-1 (NRG-1), and FGF9 [1,3,8].

## 2. Calcium and Cell Cycle Regulation

Considerable information has been amassed concerning the role of Ca^2+^ in cell cycle regulation, albeit only a fraction of it originates directly from NSCs. While the mechanisms involved are likely redundant across cell types, certain peculiarities may exist between them. This section will discuss the role of Ca^2+^ in cell cycle regulation and the involvement of effector proteins at various stages of the cell cycle within the context of NSCs.

Cytosolic Ca^2+^ concentration ([Ca^2+^]c) regulates cell proliferation, with Ca^2+^ oscillations occurring throughout the cell cycle believed to regulate this process [17,18,19]. In RGC, [Ca^2+^]c increases via the binding of morphogens and growth factors to GPCRs and RTKs or through mechanoreceptor activation [19,20,21].

Changes in [Ca^2+^]c are associated with different phases of the cell cycle and the quiescent stage. In NSCs, Ca^2+^ signaling is essential for maintaining self-renewal or promoting cell differentiation by participating in various stages of the cell cycle, including the G1, S, G2, and M phases. Ca^2+^ binds to several proteins during this process, leading to alterations in the expression of cell cycle regulatory proteins [22,23]. During the G1 phase, the increase in [Ca^2+^]c can directly activate Ca^2+^-dependent kinases, such as Ca^2+^/CaM-dependent protein kinases (CaMKs) and protein kinases C (PKC), thereby increasing the expression of cell cycle regulatory proteins, including cyclins and cyclin-dependent kinases (CDKs). Ca^2+^ affects DNA replication during the S phase by modulating the activity of enzymes involved in DNA synthesis through Ca^2+^-dependent tyrosine kinase (PYK2), CaM, the activated protein kinase (MAPK), and ERK signaling pathways [24,25]. Furthermore, in G2, Ca^2+^-dependent kinases regulate cell cycle progression by controlling the activity of cyclin B and CDK1. In the M phase, Ca^2+^ signaling is essential for spindle formation, chromosome segregation, microtubule dynamics, and the activation of proteins such as mitotic kinases [26,27].

It has been suggested that in NSCs, changes in the duration of the G1 phase dictate self-renewal or differentiation, with a shorter G1 observed during symmetric division (self-renewal) and a longer G1 observed to be NSC divide in an asymmetric and symmetric non-proliferative fashion [28,29]. The role of Ca^2+^ in symmetric or asymmetric cell division remains unclear. However, using a cyclin-dependent kinase inhibitor to prolong the cell cycle appears sufficient to induce premature neurogenesis in mouse embryos [29]. This suggests that changes in [Ca^2+^]c during the G1 phase may influence the duration of this stage, potentially affected by factors such as receptor density and type, Ca^2+^ membrane channels, and levels of Ca^2+^-dependent kinases, all of which can impact cell cycle regulatory proteins. Interestingly, elevated levels of GPCRs associated with PKC activation and Ca^2+^ mobilization have been shown to enhance neuron differentiation, hinting at the involvement of this pathway in asymmetric NSC division [25,30,31,32].

Other studies have revealed that Ca^2+^ waves are propagated through connexins in the VZ of the embryonic brain to enhance RGC proliferation [23]. Since RGCs are situated close to the ventricular lumen, they are exposed to physical and chemical factors from both the surrounding tissue and the cerebrospinal fluid (CSF) in their basal and apical regions, respectively. The ventricular system originates from an open tube in the embryonic brain that gradually closes and rapidly expands due to the accumulation of CSF [33]. Consequently, spontaneous Ca^2+^ transients may be induced through mechanoreceptor activation via CSF circulation from a caudal to a rostral direction [21]. The positive hydrostatic pressure exerted by CSF stimulates the proliferation of neuroepithelial cells and RGCs through primary cilia, which are considered specialized organelles for Ca^2+^ signaling [34,35,36]. Interestingly, only approximately 56% of RGCs obtained from E16 mice exhibit Ca^2+^ transients in response to mechanical stimulation [37], suggesting heterogeneity among RGCs [38,39].

This observation also implies that 44% of RGCs do not exhibit mechanically stimulated Ca^2+^ transients, suggesting that intracellular Ca^2+^ transients in these populations may occur through ligand–receptor activation, leading to Ca^2+^ release from storage and its subsequent entry through membrane channels. In this regard, the signal transduction pathways of GPCRs or TKs are of interest, as several morphogens and growth factors act through them. Receptor activation leads to the hydrolysis of phosphatidylinositol 4,5-biphosphate (PIP2)-producing diacylglycerol (DAG) and inositol-triphosphate (IP3); IP3 predominantly binds to its receptor in the ER, resulting in Ca^2+^ release into the cytoplasm. Subsequently, as [Ca^2+^]c increases, Ca^2+^-sensitive proteins, such as CaM [19], play a pivotal role in transducing Ca^2+^ signals within the cell, controlling the transition from one phase of the cell cycle to the next by activating CaMKs, which phosphorylate target proteins involved in the cell cycle, such as cyclin-dependent kinases, essential for cell cycle progression. Furthermore, CaM can interact with growth factor receptors, such as the EGF receptor (EGFR), to modulate their activity and regulate cell proliferation [40]. In particular, CaM intervenes in the G1-S phase boundary, the transition from the G2 phase to mitosis (M), and the anaphase–telophase transition [37,41].

The involvement of CaM in the cell cycle is further supported by the use of antagonists, such as W13, W7, and trifluoperazine, or monoclonal antibodies, which promote cell cycle arrest and the inhibition of DNA synthesis [42,43,44]. Together with CaM, Ca^2+^ stimulates the expression of genes that lead to the activation of the cyclin-dependent kinases p33cdk2 and p34cdc2, which are necessary for the progression of G2 to M and control the signaling cascade that regulates the phosphorylation of the retinoblastoma protein, controlling a restriction point in G1-phase to enter the S-phase [45,46,47]. Furthermore, CaMKs and calcineurin can activate different signal pathways, such as those mediated by the nuclear factor of activated T-cells (NFAT), nuclear kappa-light-chain enhancer of activated B cells (NFκB), and stimulate transcription through the adenosine 3′,5′-cyclic monophosphate (cAMP) response element-binding protein (CREB) affecting cell proliferation and differentiation [48,49,50].

Calcyclin, a member of the S-100 family, also known as S100 Ca^2+^-Binding Protein A6 (S100A6), is expressed in various cell types, including neuron-like and glial-like cells in the pyramidal and molecular layers of the hippocampus at midgestation, in the entorhinal cortex throughout gestation, and in the fetal occipital cortex [51,52,53]. Its expression in glia-like cells, particularly in the entorhinal cortex and the occipital cortex in fetuses, suggests an important role in neural development.

In the adult brain, S100A6 is expressed in NSC and astrocyte precursors, indicating its potential involvement in generating astrocytes in the hippocampus [54]. Other studies reported its expression in epithelial cells and its overexpression in cancer epithelial cells, suggesting a role in cell proliferation [55,56,57,58]. Additionally, its effect on fibroblasts, where its deficiency prolongs the G0/G1 phase and leads to cell cycle withdrawal, further supports its involvement in the cell cycle [59]. These findings provide insights into the potential role of calcyclin in NSC neurogenesis and gliogenesis. However, further studies are needed to understand how this protein influences the NSC division pattern for self-renewal and differentiation.

It is important to mention that Ca^2+^ not only promotes or regulates the length of the cell cycle but also plays an essential role in the degradation of certain cyclins by activating the Ca^2+^-dependent protease calpain, thereby halting cell cycle progression [60].

As mentioned above, growth factors such as the EGF and FGF stimulate NSC proliferation through receptor activation, leading to increased cytosolic [Ca^2+^]c and CaMK activation. These kinases can phosphorylate and activate transcription factors involved in cell cycle progression and proliferation, such as myocyte enhancer factor 2 (MEF2) and NFAT, and modulate the activity of CDKs and their inhibitors, thus regulating NSC progression through different cell cycle phases [23,61,62,63].

In summary, Ca^2+^ entry into NSC is crucial for cell cycle regulation, and depending on the activated pathway, NSCs will either achieve self-renewal or differentiate into specific neural lineages. Therefore, changes in the transcriptional profiles of NSCs during CNS development will determine the effects of Ca^2+^ on these cells. Consequently, we will review the expression of membrane proteins such as GPCRs, transitory receptor potential channels (TRPs), RTKs, and connexins during development and their roles in NSC proliferation.

## 3. G-Protein-Coupled Receptors and Calcium Signaling

GPCRs constitute a large family of cell surface receptors that play a crucial role in transmitting signals from the extracellular environment to the inside of the cell through the hydrolysis of heteromeric guanine nucleotide-binding proteins (G-proteins). Upon the activation of GPCR, the liberation of the alpha subunit (αi, αs or αq) determines the downstream signaling pathway [64].

G-proteins are composed of the β/γ (G_β/γ_) dimmer and the α subunits, forming a trimeric protein complex. They are classified based on their α (G_α_) subunit, which dictates their function upon receptor activation and promotes the disassembly of the trimetric protein. While the G_α_ subunits initiate the transduction signaling pathway downstream from receptor activation, the G_β/γ_ subunit regulates the activity and stability of the G_α_ subunit, thereby contributing to GPCR-mediated signaling. The G_α_ subunits are further classified into G_αs_, G_αi_, G_αq11_, and G_α12/13_ subfamilies based on their primary sequence and function. Specifically, G_αs_ and G_αi_ stimulate and inhibit cAMP synthesis, respectively, while G_α12/13_ regulates actin cytoskeletal remodeling by activating Rho GTPase, and G_αq11_ activates phospholipase C, leading to intracellular Ca^2+^ release and entry [64].

Moreover, as mentioned previously, after IP_3_R binds to its receptors in the ER and Ca^2+^ depletion occurs, conformational changes in the Ca^2+^ sensor proteins, named stromal interaction molecules (STIM) in the ER membrane, are triggered. These changes activate membrane proteins such as Ca^2+^ channels, such as ORAI (1, 2, and 3) and TRPC1, facilitating store-operated Ca^2+^ entry (SOCE) and [65,66] (Figure 2). Consequently, the increased [Ca^2+^]c serves to refill the ER and activate Ca^2+^-sensitive proteins, including CaM, CaMKs, calcyclin, calcineurine, and Ca^2+^-dependent PKCs, all of which play critical roles in NSC proliferation. For instance, CaMKII promotes cell cycle progression, entry into the S phase, and neurogenesis while also phosphorylating PKC, predominantly affecting the G0/G1 and G2 phases [67,68].

Moreover, GPCRα_q_ has been reported to influence NSC self-renewal and differentiation (Table 1). Its effect on NSC differentiation likely occurs through symmetric non-proliferative or asymmetric divisions.

## 4. Transitory Receptor Potential Channels and NSC Proliferation

The TRP ion channels comprise a large family of proteins characterized by six transmembrane domains and relatively long intracellular amino and carboxyl termini, encoded by 28 genes. These proteins form channels by arranging into tetramers (homotetramers or heterotetramers), where the S5 and S6 domains come together with the interconnecting loop to create the central pore [140,141]. Most TRPs are non-selective cation channels permeable to Na^+^, Ca^2+^, and Mg^2+^ and expressed in both excitable and non-excitable cells [142]. The mechanisms underlying the activation are poorly understood; however, they have been reported to be activated by voltage changes, receptor-operated activation, Ca^2+^-store depletion, ligands, and sensory responses to heat, cold, and pain [142,143].

The first member of the TRP superfamily was identified in Drosophila as a protein involved in phototransduction [144,145]. Since then, these proteins have been classified into seven subfamilies based on their homology to Drosophila’s photoreceptor: TRPC (canonical), TRPV (vanilloid), TRPM (melastatin), TRPP (polycystin), TRPML (mucolipin), TRPA (ankyrin), and TRPN (no mechanoreceptor potential C; nompC) [146,147]. TRP channels are implicated in several physiological processes, including cell proliferation, migration, survival, Ca^2+^ and Mg^2+^ homeostasis, neuronal growth, temperature sensation, and pain perception [142,148]. Here, we will discuss the TRP channels involved in the proliferation of NSCs.

### 4.1. TRPCs

TRPCs consist of seven members grouped into four subfamilies, all of which are activated downstream from GPCRs and intracellular Ca^2+^ release or DAG [147] (Figure 2). The first family comprises TRPC4 and TRPC5; the second includes only TRPC1; the third encompasses TRPC3, TRPC6, and TRPC7; and TRPC2 is the sole member of the fourth family; however, it is not expressed in humans as it is encoded by a pseudogene [149]. These channels are expressed early during embryo development and persist into adulthood, playing essential roles in neuronal development, including NSC proliferation, cerebellar granule cell survival, axon pathfinding, neuronal morphogenesis, and synaptogenesis [150].

Among the TRPC channels, TRPC1, TRPC3, TRPC6, and TRPC7 have been involved in cell proliferation [151,152,153,154,155,156]. Ca^2+^ mobilization through these channels occurs via store-operated and receptor-operated activation, following phospholipase C (PLC) activation downstream from G_αq/11_PCRs and RTKs, which leads to DAG and IP3 production. DAG is required for the receptor-operated activation of at least TRPC3, TRPC6, and TRPC7, while IP3 causes Ca^2+^ release from the ER and the activation of TRPC1, a process triggered by store-operated channel activation. Furthermore, TPCR4 is activated via a receptor-operated mechanism by G_αi_-PCRs (Figure 3) [157,158,159,160,161,162,163].

Strübing and colleagues (2003) reported greater expression of five TRPC proteins (TRPC1, 3, 4, 5, 6) in the embryonic compared with the adult brain. Furthermore, they proposed that TRPC1 plays an important role in forming heterotetramers composed of TRPC1/TRPC4/TRPC5 and TRPC1/TRPC3/TRPC6 [164]. Studies using smooth muscle and endothelial cells have shown that TRPC1 mediates Ca^2+^ influx induced by basic FGF (bFGF), thereby increasing cell proliferation [165,166]. Since bFGF plays an important role in cortical NSC proliferation in vitro and in vivo through FGF receptor-1 (FGFR-1) [167,168,169], it is likely that the mechanism reported in endothelial cells is also active in NSCs. Indeed, in the murine telencephalic neuroepithelium, TRPC1 and FGFR-1 are co-expressed in proliferating NSCs, and they have been coimmunoprecipitated from membrane extract preparations [164,169].

Furthermore, Ca^2+^ entrance and NSC proliferation induced by bFGF are reduced in TRPC1 knockdown NSCs [164,170], leading to cell cycle arrest in G0/G1 due to the overexpression of CaMK-II beta (Camk2b) and CDK inhibitor 2A (Cdkn2a) [171].

TRPC3 is also highly expressed in NSCs derived from mouse embryonic stem cells (mESC). Knocking out this channel in this cell type promotes impaired pluripotency, neural differentiation, and increased apoptosis, suggesting that TRPC3 activity participates in cell survival, the maintenance of pluripotency, and the transition of mESCs to NSCs through the disruption of the mitochondrial membrane potential in undifferentiated mESCs and mESCs undergoing neural differentiation. This reveals that TRPC3 is required for both early and late neural differentiation [153]. Moreover, it has recently been reported that the increased neuron differentiation of NSCs by ketamine is due to dramatic repression of the expression of TRPC3, partly by regulating the Glycogen synthase kinase 3β (GSK3β)/β-catenin pathway [172].

The above findings are relevant since medical practitioners and veterinarians use ketamine as an anesthetic during pregnancy or early after birth, which may affect brain development [173]. However, how ketamine regulates NSC differentiation and whether GSK3β participates in the ketamine-induced differentiation of NSCs is still unclear.

The participation of TRPC1 and TRPC3 in cell differentiation has also been observed after the knockdown of these channels in immortalized rat hippocampal cells (H19-7) derived from E17 Holtzman rat embryos, where cell differentiation is blocked. Another channel that is highly expressed in this cell type during proliferation and decreases dramatically upon differentiation is TRPC7 [173]. This suggests that TRPC1 and TRPC3 could be related to asymmetric division, while TRPC7 may be involved in the symmetric division of NSCs.

### 4.2. TRPVs

The TRPV family comprises six non-voltage-gated channel members named TRPV1 to TRPV6, which are further subdivided into the thermo-TRPV (TRPV1 to TRPV4) and the Ca^2+^-selective-TRPV (TRPV5 and TRPV6) subtypes [174]. In mammals, these proteins possess six ankyrins (Ank) repeat domains at the N-termini and a TRP-box in the C-termini. It has been proposed that the Ank repeat, TRPV1, and TRPV4 harbor an ATP-binding site, which might be important for channel activation and inactivation by CaM. Additionally, the TRP-box domain is believed to facilitate channel inactivation, gating, and protein tetramerization [175,176,177].

Mechanical stimulation, pH, and osmotic pressure changes are among the stimuli involved in its activation. The physiological functions of these channels vary depending on the TRPV subtype and the tissue in which they are expressed [178] (Figure 4). To our knowledge, TRPV1, TRPV5, and TRPV6 do not participate in NSC proliferation or CNS development, while TRPV2, TRPV3, and TRVP4 participate in these processes.

TRPV2 is a non-selective cation channel exhibiting Ca^2+^ permeability and highly expressed in the brain, lung, and spleen, where it contributes to Ca^2+^ homeostasis and macrophage activation [179]. It is activated by noxious heat (threshold of >52 °C), changes in osmolarity and membrane stretch, and growth factors, the latter causing PI-3K-dependent and -independent translocation to the plasma membrane [180,181].

Morgan and colleagues reported that ventral mesencephalic human NSCs exhibit high expressions of TRPV2 and TRPV3, which decline as differentiation progresses. Moreover, they suggested that these channels underline the mechanism for nearly all spontaneous Ca^2+^ activity in both proliferating and differentiating cells [182]. However, Santoni and collaborators found that the overexpression of TRPV2 in glioblastoma stem cells leads to the upregulation of GFAP and β-III tubulin levels, promotes differentiation both in vitro and in vivo, and reduces cell proliferation [183]. This suggests that the function of this channel may vary depending on the cellular type and context.

Although no effect of TRPV4 has been reported in NSCs, an interesting aspect of this channel is its association with the increased proliferation of oligodendrocyte progenitor cells (OPCs). The stimulation of rat OPCs with a selective TRPV4 agonist, GSK1016790A, induced a concentration-dependent increase in cell proliferation through Ca^2+^ influx via a PKC-dependent pathway. This effect was prevented by the TRPV4-antagonist, HC067047 [184].

### 4.3. TRPPs

TRPP is a family comprising two members: polycystin-1 (PC1) and polycystin-2 (PC2) [185]. The latter is closely related to TRPV1 and TRPV2, while PC1 is not considered to be part of the TRP family due to its molecular structure [186]. However, both PC1 and PC2 have been detected in the primary cilia of RGCs, where they appear to contribute to the planar cell polarity of late RGCs [187]. PC2 is situated in the plasma and ER membranes, where it interacts with TRPC1 and IP3R (Figure 5), thereby modulating intracellular Ca^2+^ signaling, for example, by prolonging the half-time decay of the IP3-induced Ca^2+^ transient [188,189,190].

Interestingly, these channels are expressed in mice at E12.5 in the developing cerebral cortex [191]. The knockdown of the expression of PC1 or PC2 in E13.5 primary cerebral cortical NSCs promotes an increase in proliferation and a decrease in neuron differentiation through a mechanism in which the Notch and STAT3 signaling are enhanced. The above is supported by the participation of this pathway in NSC self-renewal and blocking its differentiation [192]. Furthermore, reducing STAT3 expression leads to higher IPC production from NSCs [193]. Hence, it can be suggested that reducing PC1 and PC2 expression promotes symmetric cell division. During CNS development, these two channels may contribute to switching from symmetric to asymmetric NSC division and RGC transition to IPCs and, consequently, neuron differentiation.

## 5. Receptor Tyrosine Kinases

RTKs are cell surface receptors with intrinsic enzymatic activity regulated via ligand binding. Their molecular structure consists of an N-terminal extracellular domain for ligand recognition, a single transmembrane domain, and an intracellular kinase domain followed by a largely C-terminal tail region, which generates regulatory signals [194].

Ligand-activated RTKs stimulate many cellular processes, including protein synthesis, cell cycle progression, DNA synthesis, and cellular replication. Each RTK unleashes a specific set of responses through ligand binding that leads to the cross-phosphorylation of the RTK and the subsequent recruitment and activation of intracellular proteins, which transmit their signal to a series of targets, such as cyclins, PKC, phosphatidylinositol-3-kinase (PI-3K), GTPase, rat sarcoma virus/mitogen-activated protein kinase (Ras/MAPK), phospholipase A2 (PLA2), and transcription factors [194].

Growth factors such as PDGF, EGF, bFGF, and IGF-I promote an increase in [Ca^2+^]c [195,196,197,198]. In contrast with GPCR activation, RTK stimulation induces a long-lasting Ca^2+^ elevation that activates a specific pattern of genes, different from those regulated by the brief spikes caused by the release of Ca^2+^ from the RE and SOCE.

Several transduction pathways can be activated downstream from RTKs, and it is well established that these promote cell proliferation. However, the only pathway related to Ca^2+^ dynamics is the one associated with PLCγ activation, which promotes IP3 and DAG production (Figure 6). Recent studies suggest that PLCγ1 plays an important role in CNS development through ligand–receptor activation. However, PLCγ1 has been linked to neurite outgrowth, cell migration, axon guidance, and synaptic plasticity in vitro, and there is limited evidence linking it to NSC proliferation [199,200,201,202,203].

There is no doubt that RTK activation leads to NSC proliferation (Table 2). However, it is likely that ligand-binding activation may differentially affect cell division patterns in a spatiotemporal manner for proper CNS development. Interestingly, the PLCγ1 pathway regulates the dynamics of the actin cytoskeleton through the production of IP3 and DAG, which promote Ca^2+^ release from intracellular storage and Ca^2+^ influx, respectively [204,205].

The regulation of actin-cytoskeleton allows neuronal cells to undergo morphological changes during mitosis, cell polarity, neuronal processes extension, cell migration, synapse formation, and axon pathfinding. Interestingly, changes in the length of the basal processes of RGCs and the position of their cellular body occur during cell division, and the dynamics of actin cytoskeleton for cell proliferation are related to transducing signals from the cell surface to the nucleus, thereby regulating gene expression and cell cycle progression. Furthermore, actin filaments are essential for forming the contractile ring during cytokinesis and the regulation of cyclase-associated proteins that coordinate actin polymerization and depolymerization, which influence cell cycle progression and cell division patterns [9,206,207,208].

Despite differences in ligands, amino acid sequences, and structures, almost all RTKs can activate the same signaling pathways, which affect cell proliferation. Cellular function is likely to depend on receptor density, the coactivation of RTKs, and cross-talk with GPCRs, which selectively influence gene expression and cell function.

**Table 2 ijms-25-04073-t002:** RTKs that affect neural stem cell proliferation.

RTK	Role	Ligands/Agonists	Antagonists
EGFR (ErbB)	Increases NSC proliferation, migration, and survival in vivo and in vitro [209,210,211]. Promotes oligodendrocyte differentiation of NSCs in vivo [198,212]. The knockout mice show the atrophy of the anterior cerebral cortex in vivo [213]	EGF, TGF-α [209], Amphiregulin, Betacellulin, Heparin-binding EGF-like growth factor [214] Epiregulin, Epigen [215], Neuroregulins [211]	Cetuximab, Panitumumab [214], Trastuzumab, Pertuzumab, ABX-EGF, EMD-7200, h-R3, ICR-62, ZD1839 (Gefitinib Iressa), OSI-774 (Erlotinib), Lapatinib (GW572016, GW2016), EKB-569, AEE788, BMS-599626, AZD 9291, Dacomitinib, Afatinib, CO-1686, Neratinib, Canertinib, AC-480, AZD 8931, AST 1306 [216]
FGFR1	Maintains the self-renewal of NSCs [217]. NSC proliferation and neurogenesis in the developing cerebral cortex [169] Its deletion, together with FGFR2 and FGFR3, leads to Foxg1-positive precursors telencephalic cell death, resulting in the loss of the basal ganglia and cortex in vivo [218].	FGF-1, -2, -4, -6, -7, -8, -10, -16, -17, -18, -22 [219]	Derazantinib (ARQ087, ASP5878, AZD4547), Infigratinib (BGJ398), Debio-1347, Dovitinib [220,221], Brivanib (BMS-582664), BMS-540215, E-3810 (AL3810), NP603, LY2874455, Fisogatinib (CH518328, Debio 1347, E7090), Rogaratinib (BAY1163877), Futibatinib (TAS-120), Pemigatinib (INCB054828), Erdafitinib (JNJ-42756493) [221]
PDGFRα	The knocking down or blocking antibodies of PDGFRα suppresses the proliferation of NSCs and increases the cell death rate [222].	PDGF-A, PDGF-B, PDGF-AB, PDGF-C [223]	Dasatinib, Masitinib [224], Axitinib [225], Sorafenib, Pazopanib, Cediranib [225,226] Imatinib, Sunitinib, Nilotinib [224,225,226]
IGF-1R C	The knocking down of the receptor reduces NSC proliferation, stunts brain growth, and decreases the neuronal number [227].	IGF-II, Insulin, IGF-I (Somatomedin C) [228]	NVP-ADW742, α-IR3, JB-1 [229] NVP-AEW541 [230], MAB391, OSI-906 [231]
TrkB (Ntrk2)	Promotes NSC survival and proliferation [232] in cortical precursors in vivo and promotes proliferation and enhanced neurogenesis [233].	BDNF [232,233], NT-4, NT-3 [234], L-783,281 [235] Amitriptyline, 7,8-Dihydroxyflavone, Deoxygedunin, Paecilomycine A [236]	Larotrectinib, Entrectinib, Selitrectinib (LOXO-195), Altiratinib, DS-6051b, Lestaurtinib, Merestinib, MGCD516, PLX7486, ONO-5390556, TPX-0005, Repotrectinib [237]
TrkC (Ntrk3)	Increases neuron differentiation in vitro [238] and NSC proliferation in vitro and in vivo [233].	NT-3 [234,238], L-783,281 [235]	Larotrectinib, Entrectinib, Selitrectinib (LOXO-195), Repotrectinib, Altiratinib, Crizotinib, DS-6051b, Lestaurtinib, Merestinib, MGCD516, TSR-011, ONO-5390556, TPX-0005 [237]

Ntrk2: Neurotrophic Receptor Tyrosine Kinase 2; Ntrk3: Neurotrophic Receptor Tyrosine Kinase 3.

## 6. Gap Junctions

Connexins are proteins with four transmembrane domains named based on their predicted molecular weight. These molecules form hemichannels composed of six connexins which together form a connexon. The union between connexons in different cells forms a gap junction, allowing the direct cytoplasmic exchange in low-weight molecules and ions between adjacent cells [239]. In total, 20 and 21 types of connexins have been reported in mice and humans, respectively [240]. In the developing cerebral cortex of mice (E14–E18), Cx26, Cx36, Cx37, Cx43, and Cx45 are highly and differentially expressed [241].

In NSCs, connexins play several crucial roles, including cell–cell communication, cell synchronization, cell cycle progression, and niche maintenance. Connexins expressed during late corticogenesis in all cortical layers include Cx26 and Cx37, while those abundant in the VZ are Cx36, Cx43, and Cx45 [241]. Therefore, the latter are likely to play an important role in NSC proliferation. Moreover, their expression varies according to the phase of the cell cycle and developmental stage. The pharmacological blocking of connexins prevents DNA synthesis in NSCs [23,242,243,244,245]. Notably, reducing Cx43 translation using short hairpin technology decreases spontaneous Ca^2+^ activity and the rate of RGC proliferation, thereby reducing NSC and IP pools [23,246].

Spontaneous Ca^2+^ activity during early corticogenesis (E15) in rats propagates in clusters of RGCs within the VZ. As development progresses and neurons migrate, this Ca^2+^ pattern decreases [247]. Ca^2+^ waves occur through gap junctions, initiated by ATP binding to the GPCR subtype P2Y1, coupled to a G_αq/11_ protein. This leads to the activation of PLCγ, IP3 synthesis, and intracellular Ca^2+^ release due to IP3R activation in the ER membrane [23,248,249].

As mentioned above, Cx43 is one of the most prominent connexins implicated in NSC proliferation (Figure 7). Cx43 forms gap junction channels that directly connect the cytoplasm of adjacent cells, essential for coordinating NSC proliferation within their niche. The pharmacological blocking of gap junctions or knockdown Cx43 stunts the nuclear migration of the S/G2-phase cells in the upper strata of the VZ, while the knockout of this connexin disorganized the VZ/SVZ and promoted deficiencies in neuronal migration in mice [245,250].

Furthermore, Cx43 facilitates the integration of environmental signals within the NSC niche by allowing NSCs to communicate with each other and responding to growth factors, neurotransmitters, and other cues influencing proliferation. Although Cx43 appears particularly important for NSC proliferation, other connexins, such as Cx30 and Cx45, may also contribute to cell–cell communication within NSC populations [251]. However, their specific roles in regulating NSC proliferation are less characterized than Cx43.

**Figure 7 ijms-25-04073-f007:**
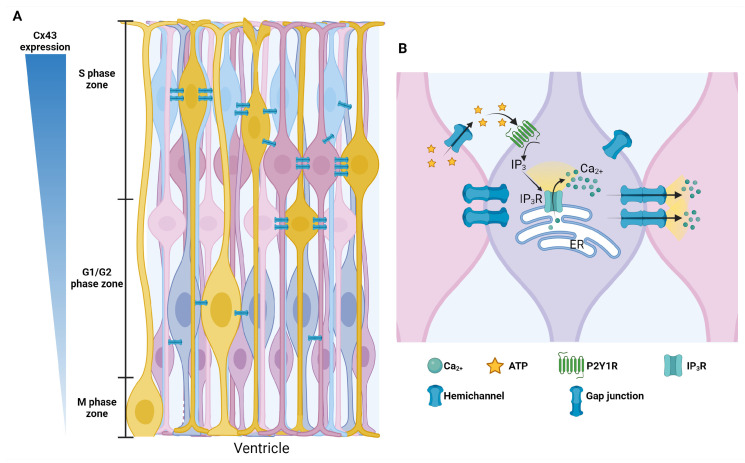
Radial glia cell cycle and conexin-43. (**A**) Scheme showing the level of expression of Cx43 during the cell cycle and gap junction. During the S Phase, Cx43 is highly expressed, and RGCs are coupled; as the cell cycle progresses, Cx43 decreases its expression, and RGCs are less coupled, while in the M phase, cells are uncoupled. (**B**) In the S phase, radial glia initiates Ca^2+^ waves by releasing ATP, which binds to P2Y1 receptors in the membrane of adjacent cells, inducing IP3 Ca^2+^ release from the endoplasmic reticulum (ER). Created using BioRender.com based on [23,242,252].

## 7. Conclusions

It is evident that NSC proliferation and [Ca^2+^]c levels are closely related. Therefore, the regulation of cell proliferation and differentiation during brain development must be finely tuned to ensure correct cytoarchitecture and function development. As development progresses, molecules involved in Ca^2+^ dynamics exhibit changes in expression, impacting molecule–cell interactions and downstream signaling pathways in a temporally and spatially different manner. Furthermore, studying NSCs in vivo and in vitro presents challenges due to the diversity of NSC populations coexisting with progenitor-committed and differentiated cells as development progresses.

By providing an overview of the role of Ca^2+^ signaling in NSC proliferation and differentiation, we aim to deepen our understanding of the intricate relationship between Ca^2+^ signaling and its dynamics in NSCs, particularly focusing on proliferation and differentiation process. Through in-depth exploration of the role of RGCs and the impact of Ca^2+^ fluctuations on cell cycle regulation, we underscore the complex mechanisms governing CNS development. By synthesizing findings from several studies, we emphasize the significance of Ca^2+^ signaling pathways in orchestrating the fates of NCSs, with potential implications for understanding neurodevelopmental processes and devising therapeutic interventions for neurological disorders originating in fetal development. This review underscores the expanding knowledge in this field and the importance of further research to unravel the nuances of Ca^2+^-mediated signaling in NSC biology.

## Figures and Tables

**Figure 1 ijms-25-04073-f001:**
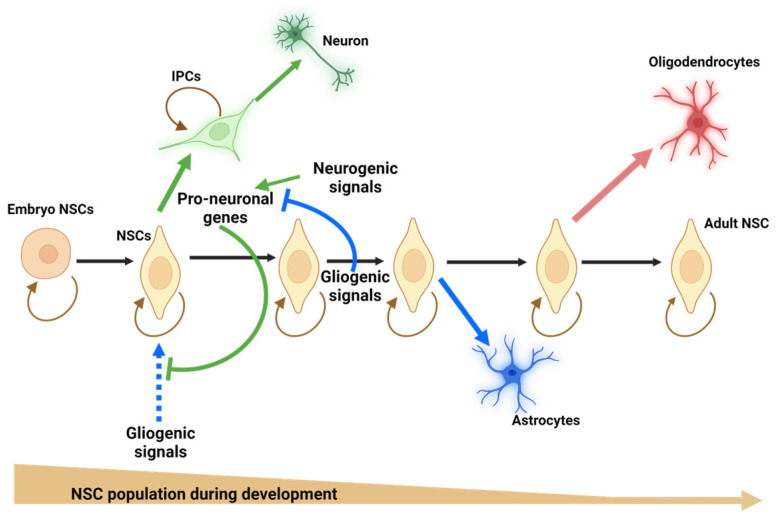
Scheme of the cellular phenotype generated during development from a heterogeneous population of neural stem cells. NSCs proliferate symmetrically (
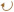
) to increase their pool or asymmetrically (black) to give rise, directly or indirectly, in a time-dependent manner, to neurons (green arrows), astrocytes (blue arrows), and oligodendrocytes (red arrow). NSCs = neural stem cells; IPCs = intermediate progenitor cells. Blue doted arrow = repressed gliogenic signals; Blunt green arrow = pro-neuronal genes repressing gliogenic signals; Blunt blue arrow = gliogenic signals repressing neuronal genes. Created using BioRender.com.

**Figure 2 ijms-25-04073-f002:**
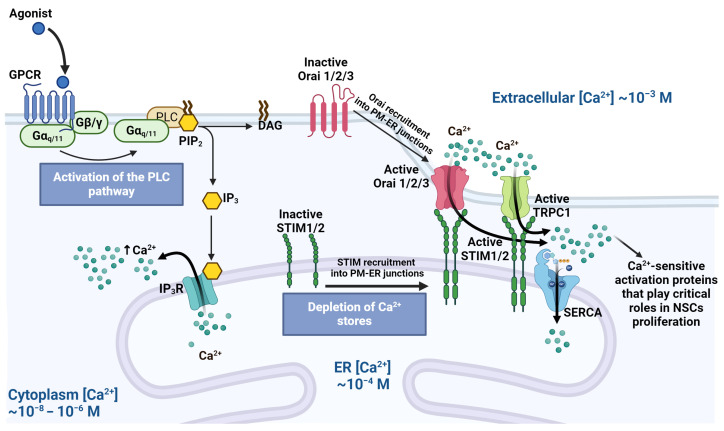
Calcium entry stimulated via GPCR activation in neural stem cells. G_αq_PCR is activated through ligand binding promoting PLC activation, Ca^2+^ is released from the endoplasmic reticulum (ER), Ca^2+^ depletion in the ER is sensed by the stromal interaction molecules (STIM) and its conformational change to move and contact Ca^2+^ channels (ORAI and TRPC1) in the cell membrane to promote Ca^2+^ entrance for ER refill through sarcoendoplasmic reticulum Ca^2+^ ATPase (SERCA) and bind to proteins with roles in cell proliferation. Created using BioRender.com.

**Figure 3 ijms-25-04073-f003:**
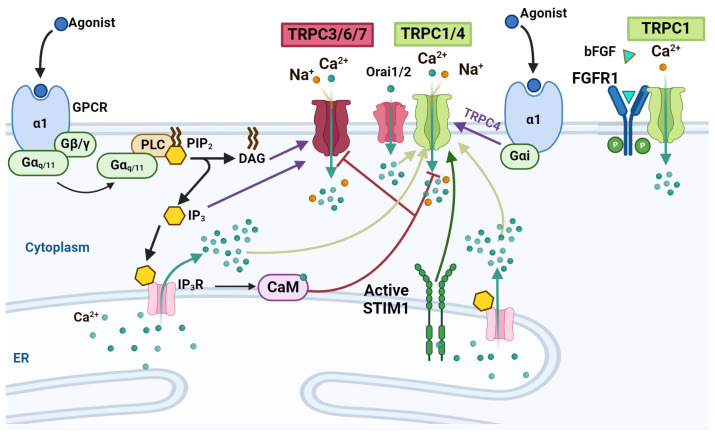
TRPCs in neural stem cells. Scheme showing the two pathways for TRPCs activation: the store-operated (light green arrows) and receptor-operated activation (purple arrows) pathways. Black arrows = agonist GPCR pathways involving G_αq/11_ and downstream signaling pathway for Ca^2+^ ER release; green arrows in the middle of TRPCs and IP_3_R = extracellular/intracellular Ca^2+^ entrance and release from de ER, respectively. And, Blunt brown arrow = negative regulation of CaM on TRPCs. Created using BioRender.com.

**Figure 4 ijms-25-04073-f004:**
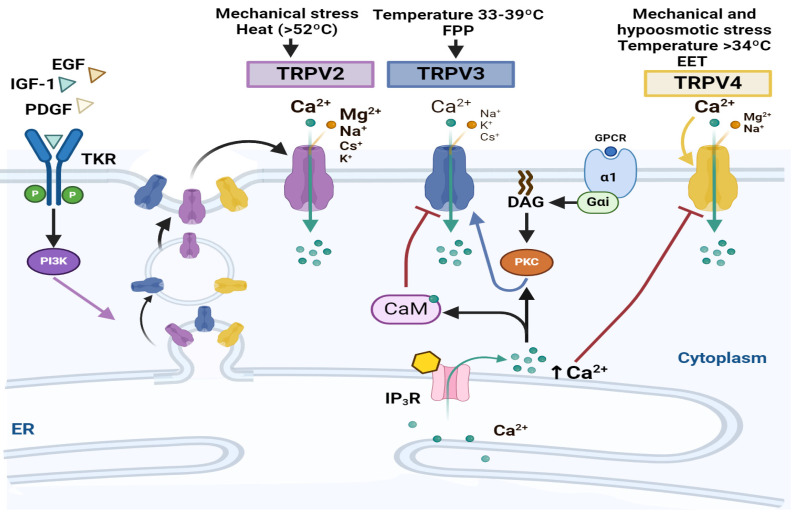
TRPVs and neural stem cell proliferation. The Scheme shows the TRPVs’ activation signals and related pathways for neural stem cell proliferation. Purple arrow = PI3K participation in TRPV cell membrane localization. Once in the membrane (black arrow in the left) TRPVs can be activated promoting Ca^2+^ entrance (green arrows). TRPV3 and 4 are negatively regulated by CaM (blunt brown arrow) and can be activated by PKC (blue arrow) through GPCR DAG and Ca^2+^ release (black arrows in the right). Yellow arrow = extracellular Ca^2+^ influence TRPV4 activity. Created using BioRender.com.

**Figure 5 ijms-25-04073-f005:**
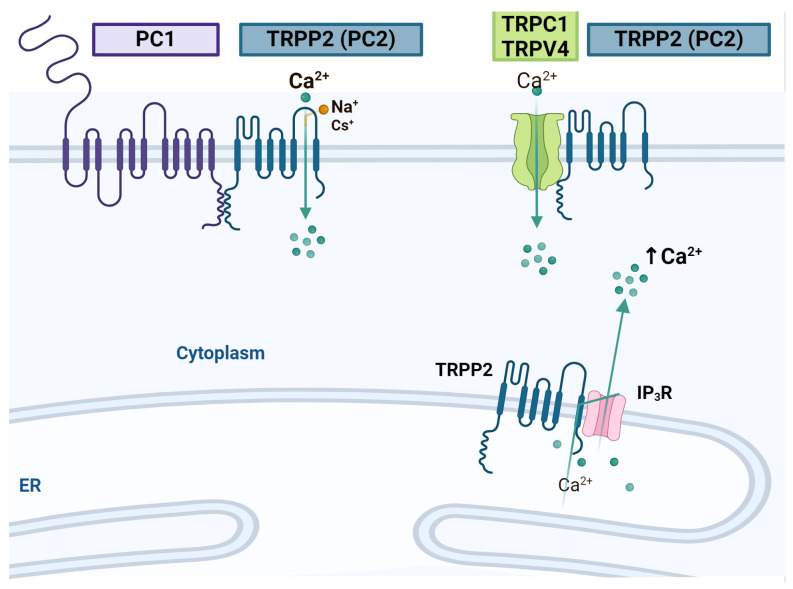
TRPPs. Scheme showing the interaction of PC2 with other channels and receptors in the plasma membrane and the endoplasmic reticulum membrane (ER). Created using BioRender.com.

**Figure 6 ijms-25-04073-f006:**
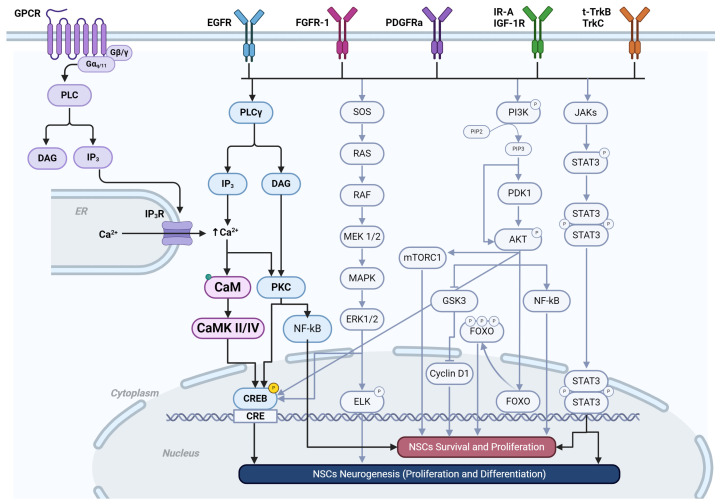
RTK pathways. Scheme showing the pathways activated by RTKs. The pathways in the black arrows are the PLCγs and the cross-taking with GPCR receptors, both promoting an increase in [Ca^2+^]_i_. Created using BioRender.com.

**Table 1 ijms-25-04073-t001:** Gαq-coupled receptors that affect cell proliferation.

Receptor	Function	Ligand	Agonists	Antagonist
5-HT2AR (serotonin 2A receptor)	Promotes NSC proliferation ex vivo, in vitro, and in vivo [69] Its antagonism increases NSC differentiation in vitro [70]	Serotonin	LSD [71], DOI [72], 25CN-NBOH [73], Mescaline [74], Pimavanserin [75], S 16924 [76]	Spiperone [71,77], compound 3b [PMID: 28943244] [78], M100,907 (volinanserin), Pirenperone [71], (−)-MBP (meta-bromo-phenylisopropylamine) [79], Eplivanserin hemi fumarate (SR-46349B) [80], Sarpogrelate, Naftidrofuryl [81], Risperidone, Pipamperone [82], Olanzapine, Ketanserin, Clozapine, Zotepine, Ziprasidone, ORG5222, Tiaspirone, Ocaperidone [83], Ketanserin, Altanserin [84]
5-HT2CR (serotonin 2C receptor)	Its antagonism increases NSC differentiation in vitro [70].	Serotonin	LSD [71], DOI [72], Mescaline [74], (−)-MBP (meta-bromo-phenylisopropylamine) [79], RO 60-0175 [85], Lorcaserin [86], MK-212, WAY-161503 [87]	Spiperone [71], RS-102221[87], Mianserin, 1-NP, ICI 169,369, LY 53857 [84], SB206553, SB242084 [88], Nefazodone, Mirtazapine [89], Ritanserin, Mesulergine [90]
M1R (muscarinic type 1 receptor)	Promotes NSC differentiation in vitro [91,92].	Acetylcholine	AF102B, AF150 (S), AF267B [93], Xanomeline [94], Sabcomeline [95], AC-42, TBPB, N-desmethylclozapine [96], Pirenzepine, Carbachol [97], 77-LH-28-1 [98]	Pirenzepin [99], Telenzepine [100], Biperiden [101], Clemastine [102]
H_1_R (Histamine type 1 receptor)	Increases NSC neuron differentiation in vitro. Its antagonism reduces neurogenesis in vivo [32].	Histamine	PEA [103], Beta-histine [104], Histaprodifen [105], 2-TEA [106]	Diphenhydramine, Pyrilamine [106], Chlorpheniramine, Mepyramine [107], Promethazine [108], Cetirizine [109], Hydroxyzine [110], Clemastine [111], Loratadine, Desloratadine [112], Fexofenadine, Levocetirizine [113], Azelastine, Acrivastine, Astemizole, Ebastine, Fexodenadine, Ketotifen, Mizolastine, Terfenadine [114]
ADRα-1AR (adrenergic α1-receptor)	Increases NSC proliferation in vitro [115].	Norepinephrine (noradrenaline) Epinephrine (adrenaline)	Phenylephrine [116], Methoxamine [117], Metaraminol, Midodrine, Xylometazoline, Oxymetazoline, Naphazoline, Tetrahydrozoline [118], Clonidine, Cirazoline, Sgd 101/75, St 587, Amidephrine, SKF89748, SDZ NVI 085, SK&F 102652, ST-1059, A-61603, A-204176, NS-49, ABT-866, BMY 7378 [119]	Prazosin [116], Terazosin [117], Tamsulosin [120], Phentolamine [121], Doxazosin [122], Alfuzosin [123]
CCK1R (Cholecystokinin type 1 receptor)	Increases NSC proliferation and differentiation into neurons in vitro [124].	CCK-8	A-71623 [125], SR-146131 [126], FPL 14294, AR-R 15849, A-71623, PD149164, PD170292, PD151932, GI 18177 [127]	SR 27897 [126], L-364,718, Devazepide [128], Dexloxiglumide, Lorglumide, Proglumide [129], and MK 329 (devazepide) [127]
CaSR (Ca^2+^-sensing receptor)	NSCs differentiate to the oligodendrocyte [130].	Ca^2+^, Mg^2+^, L-tryptophan [130], spermine [131]	Neomycin [132], Vitamin D, Velcalcetide [133]	Calcilytics, Phosphate [134], Ronacaleret [135], 2-methyl-3-phenethyl-3*H*-pyrimidin-4-one [136], compound (S)-3h [PMID: 15686947] [137], compound 17 [PMID: 15300839] [138], 1-arylmethylpyrrolidin-2-yl ethanol amine [139]

## Data Availability

No new data were created or analyzed in this study. Data sharing is not applicable to this article.

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
