# Peer review of "Calcium and Neural Stem Cell Proliferation"

_ijms, 2024, doi:10.3390/ijms25074073_

Round 1
Reviewer 1 Report
Comments and Suggestions for Authors
This is a review article relating to the role of intracellular calcium on neural stem cell (NSC) proliferation and differentiation. The manuscript is comprehensively summarized about this topic. Overall, this is interesting, informative, and useful for the researchers in the fields of stem cell and brain science. Unfortunately, there are considerable careless mistakes throughout the manuscript. The authors should recheck the manuscript carefully.
1. Page 1, line 26 and 34, RGC is duplicated.
2. Page 2, in Figure 1, Does IP mean IPC? There is a blank parenthesis ( ) in the figure legend.
3. Page 2, line 67, citations from 15 to 19 do not relate with the role of Ca2+ in NSC proliferation. These just relate to proliferation. Please choose more proper one.
4. Page 4, line 169, glycogenesis may be wrong. It may be gliogenesis.
5. Page 5, line 203 to 218, this paragraph is redundant with line 144 to 159 on page 4. Please reconsider these parts. Line 212, calcynin may be calcyclin. Figure 2 is missing, also.
6. Page 7, line 228, 28 channel subunit genes is better to be replaced with 28 homolog genes or 28 TRP channel genes.
7. Page 7, line 246, G-protein receptors may be G-protein coupled receptors. Figure 2 may be Figure 3?
8. Page 7, line 257, Gaq/11-protei-coupled is Gaq/11-protein-coupled.
9. Page 8, line 287 to 290, this sentence may be wrong. The mitochondrial membrane potential may be critical for transition to NSCs from mESCs. Is it correct?
10. Page 10, line 352, mouse embryos is better than embryo mice.
11. Page 9, line 295, the authors mention that GSK3b is a TRK but this is wrong. The authors use TRK and TKR through the manuscript. If it means receptor tyrosine kinase, it is better to use RTK.
12. Page 9, line 296 and 297, the authors add 3 citations, but they are not assigned.
13. Page 9, line 298, please fix “is used as used” properly.
14. Page 11, line 284, Ras activation protein (GAP) may be GTPase activating protein (GAP) or Ras-GTPase activating protein (Ras-GAP). Ras mitogen-activated protein kinase (MAPK) may also be better to be used mitogen-activated protein kinase (MAPK) or Ras/mitogen-activated protein kinase (Ras/MAPK)?
15. Page 14, line 441, NSCs instead of NSCS.
16. Page 15, line 488, great instead of grate?
17. Superscript is not sometimes used for Ca2+. Please check throughout the manuscript.
Comments on the Quality of English LanguageThere are a couple of careless mistakes.
Author Response
We have attached a PDF file with a point -by-point answers

Reviewer 2 Report
Comments and Suggestions for Authors
In this work the authors are providing an extensive overview of the role of calcium during CNS development, affecting neural stem cell proliferation. The amount of collected literature is quite impressive (over 250 cited references, in particular the Table 1 and 2 represent extensive list of agonists and antagonist of G-protein coupled and tyrosine kinase receptors, affecting proliferation and very useful for pharmacological approaches).
However, there are some important issues that need to be addressed:
- The majority of references are quite old (70’s-90’s) and it is not clear what is the innovative contribution of this review that could not be found in the textbooks. It is correct to recognize pioneer works, but it is important to highlight recent evidence as well
- Figure 1 contains terms and shows processes not described in the text (e.g. neuro- and gliogenic signals, genes etc were not mentioned at that point and the link with calcium is not clear; IP is not defined in the figure legend while in the main text the abbreviation “IPCs” was used; finally, in the figure legend (Line 64 there are empty brackets)
- Figure 2 is missing.
- Author contribution contains unfinished sentence.
Minor points:
Line 34: remove “RG cells” and used RGCs since this is defined already (Line 26).
Line 77, 99: check here and for the rest of the manuscript that Ca2+ is written correctly.
Line 85: TKR abbreviation has been already defined (Line 41), while for CaMKs and PKCs (Line 92) full name should be provided at first mention. For CaMKs this is only done at Line 138 while at Line 177 again both full name and abbreviation is provided
Line 105, 180 etc: use abbreviated form for CDK
Line 111: G-protein (instead of G-Protein); abbreviation (GDPR) should be also provided here since later in the text is used etc etc
Line 179: abbreviations (MEF2 and NFAT, respectively) should be put in brackets
Please make sure that for the rest of the manuscript full names and abbreviations of proteins are provided at first mention and use then abbreviated form for the rest of the MS.
Line 192: “heteromeric guanine nucleotide-binding proteins G Protein hydrolysis” – this part of the sentence is not clear
Line 433: I suggest using either “Gap junctions” or “Connexins” for the title of the Section 5
Line 457: Remove “Interestingly” unless Cx43 is significantly different from other connexins.
Comments on the Quality of English Language
Line 15: The sentence should be rephrased to be more clear and less obvious (“symmetric and asymmetric division for proliferation and differentiation”).
Line 16: Add “It” at the beginning of the sentence.
Line 18: The sentence “Here, we draw on various studies and research papers to provide a comprehensive overview.” is obvious and therefore unnecessary.
In line with this, a lot of parts of the manuscript contain sentences not clearly written and therefore extensive English language editing is required.
Other examples of unclear sentences:
- Line 47: “perceived or sensed by several proteins, such as buffers” ??? (to sense or to perceive are synonyms; what kind of buffers?)
- “self-renewal” should be used instead of “renew” (lines 27, 88,102,184 etc)
Line 32-33: add “respectively” after VZ and SVZ
Line 69-70: the fluctuations should be isolated, paired or clustered (instead of isolate, pair and cluster)
Line 485: Remove “in this process” at the end of the sentence to avoid repeating.; The next sentence should be corrected: …as development progresses, the molecules involved….
Line 490-501: NSC (instead of full name) should be used for consistency with the rest of the text.
Author Response
We have attached a PDF file with point-by-point answers

Reviewer 3 Report
Comments and Suggestions for Authors
In the manuscript “Calcium and Neural Stem Cell Proliferation” Díaz-Piña et al review the role of Calcium in neural stem cell proliferation and differentiation, especially during development.
The structure of the review is simple to follow, but full of details and a comprehensive inclusion of several mechanisms involved in calcium regulation and signaling pathways.
I found the review interesting, easy to read and understand, and very enlightening. I think it can be useful for the readers of this journal and merits its publication as it is. As a reviewer, I just want to raise some minor questions:
1) In page 5 there are a reference to figure 2, but this figure seems to be missing.
I honestly did not find any issue that was worth reporting, but I feel compelled, as a reviewer, to indicate some additional issues. Please assess whether these should be considered (it might not be necessary, and no reply is a good reply for these two).
2) The approach to do this review is not clear. The authors do not indicate that the review is systematic, nor explain why some of the works are included or not. Although the expertise of some of the authors support a more freely crafted manuscript, it might be beneficial to include an explanation of the approach taken to prepare the manuscript.
3) In lines 438-439, the authors wrote “Twenty and 21 types of connexins have been reported in mice and humans” I understand that 20 types are reported for mice and 21 for humans, and I understand that you cannot start a sentence with a number, but it resulted confusing for me in the first read
Author Response

(The authors gave the same response as above.)

Round 2
Reviewer 2 Report
Comments and Suggestions for Authors
The scientific content of the revised version of the manuscript is now acceptable. However, additional editing of English language is still required:
- - new (revised) version of abstract still contains unclear sentences:
Line 12: „a comprehensive understanding“ (instead of comprenhensively)
Line 15-16: explain the meaning of „...a deeper investigation...is warranted(?)“
Line 18-20: sentence requires rephrasing (unclear if this is a question or statement): „And how these might play a key role in balancing cell proliferation for self-renewal or promoting differentiation, processes finely regulated in a time-dependent manner throughout brain development, influenced by extrinsic and intrinsic factors that directly or indirectly modulate calcium dynamics.”
- The correct forms should be „self-renewal capacity“ or „capacity to self-renew“ – this was not corrected by the authors (see comment no. 3 of the Answer to reviewer)
- Line 89: remove „And,“ at the beginning of the sentence
Comments on the Quality of English LanguagePlease see my previous comments.
Author Response
We appreciate the reviewer’s comments, which have helped improve our manuscript. The changes made to this version of the manuscript are now highlighted in blue. A list of changes according to the comments is listed below, and we hope you will find the new version suitable for publication.
Answers to Reviewer 2 (Round 2)
The scientific content of the revised version of the manuscript is now acceptable. However, additional editing of English language is still required:
- new (revised) version of abstract still contains unclear sentences:
Line 12: „a comprehensive understanding“ (instead of comprenhensively)
Line 15-16: explain the meaning of „...a deeper investigation...is warranted(?)“
Line 18-20: sentence requires rephrasing (unclear if this is a question or statement): „ And how these might play a key role in balancing cell proliferation for self-renewal or promoting differentiation, processes finely regulated in a time-dependent manner throughout brain development, influenced by extrinsic and intrinsic factors that directly or indirectly modulate calcium dynamics.”
Answer: After English editing the abstract and the manuscripts have been rephrased and corrected to enhance clarity and maintain the integrity and accuracy of the content.
- The correct forms should be “self-renewal capacity“ or "capacity to self-renew“ – this was not corrected by the authors (see comment no. 3 of the Answer to reviewer)
Answer: the comment was: “self-renewal” should be used instead of “renew” (lines 27, 88,102,184 etc)
Our answer in round 1: Throughout the manuscript, self-renew was used instead of renew and auto-renew.
Sorry about this. We replace self-renew with self-renewal.
- Line 90: remove „ And “ at the beginning of the sentence
Answer: And was removed, and the sentence starts with “Oligodendrocyte ……” This is on page 2, line 90.